# The 3D structure of lipidic fibrils of α-synuclein

Benedikt Frieg [1,7], Leif Antonschmidt [2,7], Christian Dienemann [3], James A. Geraets [1], Eszter E. Najbauer[2], Dirk Matthes[4], Bert L. de Groot [4], Loren B. Andreas [2], Stefan Becker [2], Christian Griesinger [2,5] ✉ & Gunnar F. Schröder [1,6] ✉

α-synuclein misfolding and aggregation into fibrils is a common feature of α-synucleinopathies, such as Parkinson's disease, in which α-synuclein fibrils are a characteristic hallmark of neuronal inclusions called Lewy bodies. Studies on the composition of Lewy bodies extracted postmortem from brain tissue of Parkinson's patients revealed that lipids and membranous organelles are also a significant component. Interactions between α-synuclein and lipids have been previously identified as relevant for Parkinson's disease pathology, however molecular insights into their interactions have remained elusive. Here we present cryo-electron microscopy structures of six α-synuclein fibrils in complex with lipids, revealing specific lipid-fibril interactions. We observe that phospholipids promote an alternative protofilament fold, mediate an unusual arrangement of protofilaments, and fill the central cavities of the fibrils. Together with our previous studies, these structures also indicate a mechanism for fibril-induced lipid extraction, which is likely to be involved in the development of α-synucleinopathies. Specifically, one potential mechanism for the cellular toxicity is the disruption of intracellular vesicles mediated by fibrils and oligomers, and therefore the modulation of these interactions may provide a promising strategy for future therapeutic interventions.

α-synucleinopathies, including Parkinson's disease (PD), multiple system atrophy, and dementia with Lewy bodies (DLB), are the second most common class of neurodegenerative diseases after Alzheimer's disease[1,2]. While the pathological causes for α-synucleinopathies are largely unknown, a common feature of these diseases is the presence of fibrillar aggregates of α-synuclein (αSyn)[3–5].

The biological function of αSyn is not well understood. However, it is known that αSyn transiently binds to lipid membranes and free fatty acids via its N-terminal domain[6], and has been connected to vesicle-trafficking, clustering of synaptic vesicles, and neurotransmitter release[7]. αSyn-related dysregulation of lipid homeostasis has been discussed as relevant for the development of PD pathology for decades[8,9]. A pathological hallmark for PD is the presence of large neuronal inclusions called Lewy bodies (LB), which have been identified to contain αSyn fibrils and lipids[10–15].

Fibrillization of αSyn is strongly enhanced in the presence of lipid vesicles compared to fibrillization in bulk solution[16], emphasizing the key role that membrane interactions may have in aggregation. Genetic

[1]Institute of Biological Information Processing (IBI-7: Structural Biochemistry) and JuStruct: Jülich Center for Structural Biology, Forschungszentrum Jülich, Jülich, Germany. [2]Department of NMR-Based Structural Biology, Max Planck Institute for Multidisciplinary Sciences, Göttingen, Germany. [3]Department of Molecular Biology, Max Planck Institute for Multidisciplinary Sciences, Göttingen, Germany. [4]Department of Theoretical and Computational Biophysics, Max Planck Institute for Multidisciplinary Sciences, Göttingen, Germany. [5]Cluster of Excellence "Multiscale Bioimaging: From Molecular Machines to Networks of Excitable Cells" (MBExC), University of Göttingen, Göttingen, Germany. [6]Physics Department, Heinrich Heine University Düsseldorf, Düsseldorf, Germany. [7]These authors contributed equally: Benedikt Frieg, Leif Antonschmidt. ✉e-mail: cigr@mpinat.mpg.de; gu.schroeder@fz-juelich.de

studies have revealed that mutations in several vesicle-trafficking-related genes can lead to the development of PD[17]. Lipid extraction by the assembly of αSyn oligomers and fibrils at the membrane surface eventually leading to membrane disruption has been proposed as a potential mechanism for toxicity[18,19]. However, it is yet unclear whether αSyn aggregates are responsible for impairing lipid homeostasis or whether it is aberrant vesicle-trafficking that causes αSyn aggregation. Hence, there are two competing models for synucleinopathies; should they be considered "lipid-induced proteinopathies" or rather "protein-induced lipidopathies"[9,13]? In both cases targeting the lipid-fibril interactions is a promising therapeutic strategy.

Detail of specific interactions of lipids with αSyn-fibrils provides crucial insight into understanding the role of lipids in PD, however, very little has been determined to date. To address this, we have elucidated the structures of six αSyn fibril-lipid complexes with cryo-electron microscopy (cryo-EM), and reveal the lipid-fibril interactions using molecular dynamics (MD) simulations together with solid-state Nuclear Magnetic Resonance (ssNMR) spectroscopy[20].

## Results and discussion

### Variation of the aggregation protocol leads to different polymorph populations

De novo aggregation in the presence of small unilamellar vesicles (SUVs) at a 5:1 lipid to protein ratio was induced by sonication under protein misfolding cyclic amplification conditions and completed under gentle orbital shaking to elongate the fibrils[20]. SUVs consisted of a 1:1 mixture of 1-palmitoyl-2-oleoyl-sn-glycero-3-phosphate (POPA) and 1-palmitoyl-2-oleoyl-sn-glycero-3-phosphocholine (POPC) as a simplification of negatively charged synaptic vesicles[21] to recapitulate the established binding of monomeric αSyn to lipids via its N-terminus[22,23]. In agreement with previous studies we observed significantly reduced lag-times in the presence of these phospholipids[24]. We confirmed the presence of αSyn fibrils by cryo-EM screening and selected three preparations of αSyn fibrils for which independent image datasets were collected. Extensive classification during 3D reconstruction revealed three dominant protofilament folds (L1, L2, and L3) that form in total six different fibrils by different quaternary arrangement. Short sonication periods favor fibrils of the L1 fold (Fig. 1a–e), while extensive sonication is needed to yield larger populations of L2 and L3 fibrils (Fig. 2a–f, Supplementary Figs. 1, 2).

### L1 αSyn fibrils share an alternative, lipid-stabilized protofilament fold

The L1 fibrils were determined to a resolution of 3.2 Å for L1A and 3.0 Å for L1B and L1C, allowing to accurately model residues M1-Q99 (Fig. 1b–d, Supplementary Fig. 3a–c). Each monomer in the L1 protofilament comprises ten β-strands (β1 to β10) with nine connecting loops (Fig. 1e). Strands β2 and β8 form the tightly packed core, stabilized by a predominant hydrophobic steric zipper[25]. The lipidic L1 fold reveals minor similarities to previously resolved structures of αSyn in the absence of phospholipids (Supplementary Fig. 4a). In detail, only the fold of the L1 segment V52–T72 is found in the protofilament fold of wild type and Y39 phosphorylated αSyn (Supplementary Fig. 4b, c). This discrepancy with previously resolved structures is probably related to the presence of phospholipids during αSyn aggregation. While the previously determined structures are characterized by a predominantly hydrophobic core, in the L1 fold a surprisingly large number of hydrophobic residues are found on the surface (Supplementary Fig. 4d). However, these "solvent-exposed" areas are decorated with non-proteinaceous densities (Fig. 1f–h), corresponding to surface-bound phospholipids (for details, see below). Hence, the phospholipids may shield, at least to some extent, the hydrophobic amino acids on the fibril surface for direct interactions with water during αSyn aggregation, which then leads to the lipid-mediated L1 fold.

While the L1A fibril consists of a single L1 protofilament (Fig. 1b), L1B and L1C fibrils are composed of two identical and intertwined L1 protofilaments (Fig. 1c, d). The L1B and L1C fibrils differ in their protofilament interfaces. In the L1B fibril, both protofilaments are related by an approximate $2_1$ screw symmetry and the protofilaments are tilted by ~37° to each other (Supplementary Fig. 5). Protofilament dimerization, mediated by hydrophobic interactions between residues M1 and V40 as well as G41, accommodates a wide cavity in the protofilament interface. The protofilaments in the L1C fibril, on the other hand, are related by $C_2$ symmetry and ionic interactions between residues K43, K45, and E57 form the inter-protofilament interface.

### L2 and L3 fibrils reveal alternative lipid-stabilized quaternary structures of common protofilament folds

The L2A fibril was determined to a resolution of 2.7 Å, 3.1 Å for L2B, and 2.8 Å for L3A (Fig. 2b–d, Supplementary Fig. 3d–f). The L2 fold is similar but not identical to αSyn "polymorph 2" (PDB ID: 6SST), first reported by Guerrero-Ferreira et al.[26] (Fig. 2e, Supplementary Fig. 6a, b). The main structural difference between the folds is a shift of strand β5 by about 10 Å with respect to strands β8 and β1. The L3 fold is similar to the fold determined for the E46K variant (PDB ID: 6UFR)[27] (Fig. 2f, Supplementary Fig. 6c). Here, the main structural differences are shifts of strands β4-β7 relative to their position in 6UFR and the presence of the N-terminal $_{14}$GVVAAA$_{19}$, which forms another β-strand neighboring β7.

The L2A fibril is composed of three identical L2 protofilaments related by a $C_3$ symmetry. Interestingly, the three protofilaments are separated by ~20 Å and thus show no direct protein-protein contacts (Fig. 2b). In the L2B fibril, two identical but asymmetrically arranged L2 protofilaments form the mature fibril and the L3A fibril reveals a similar protofilament orientation (Fig. 2c–f). In both L2B and L3A fibrils, the helical axes of the two protofilaments point in opposite directions, which leads to an identical structure and elongation kinetics of both fibril ends. These structures belong to a symmetry class of amyloid fibrils which has been postulated[28] but, so far, not been observed experimentally.

### Phospholipids bind to the fibril surface, stabilizing the alternative protofilament fold and the alternative quaternary arrangements

For all fibril structures, the cross-section cryo-EM maps reveal additional ring- and rod-shaped densities at the fibril surface (Figs. 1f–h, 2g–i, 3a), which together are reminiscent of the cross-section of phospholipid micelles. In addition, previous ssNMR experiments identified residues of αSyn that bind to phospholipids[20], and most of those residues are neighboring these ring- and rod-shaped densities (Fig. 3a, b, d, f). We, therefore, assign the micelle-like cross-section densities to phospholipids bound to the fibril surfaces and refer to these protein-lipid aggregates as lipidic fibrils.

To validate this interpretation of the extra densities, we performed MD simulations of free lipid diffusion in the presence of the αSyn fibril structures determined here. Comparable simulations have successfully identified binding sites for biomarkers on other amyloid fibrils[29,30]. The initially randomly distributed lipid molecules associate towards micelle-like aggregates (Supplementary Movie S1) and subsequently bind to predominantly hydrophobic areas on the fibril surface (Supplementary Fig. 7). The conversion of the SUVs used for the preparation of the lipidic fibrils to such small lipid aggregates upon fibril formation was confirmed by $^{31}$P ssNMR (Supplementary Fig. 8). We calculated average density grids, showing the probability distribution of the lipids relative to the αSyn fibrils, averaged over multiple independent MD trajectories. Comparison to the cryo-EM cross-sections shows that the average lipid density from MD simulations almost perfectly matches the micelle-like densities in the cryo-EM cross-sections (Fig. 3c, e, g). Additionally, the averaged lipid density

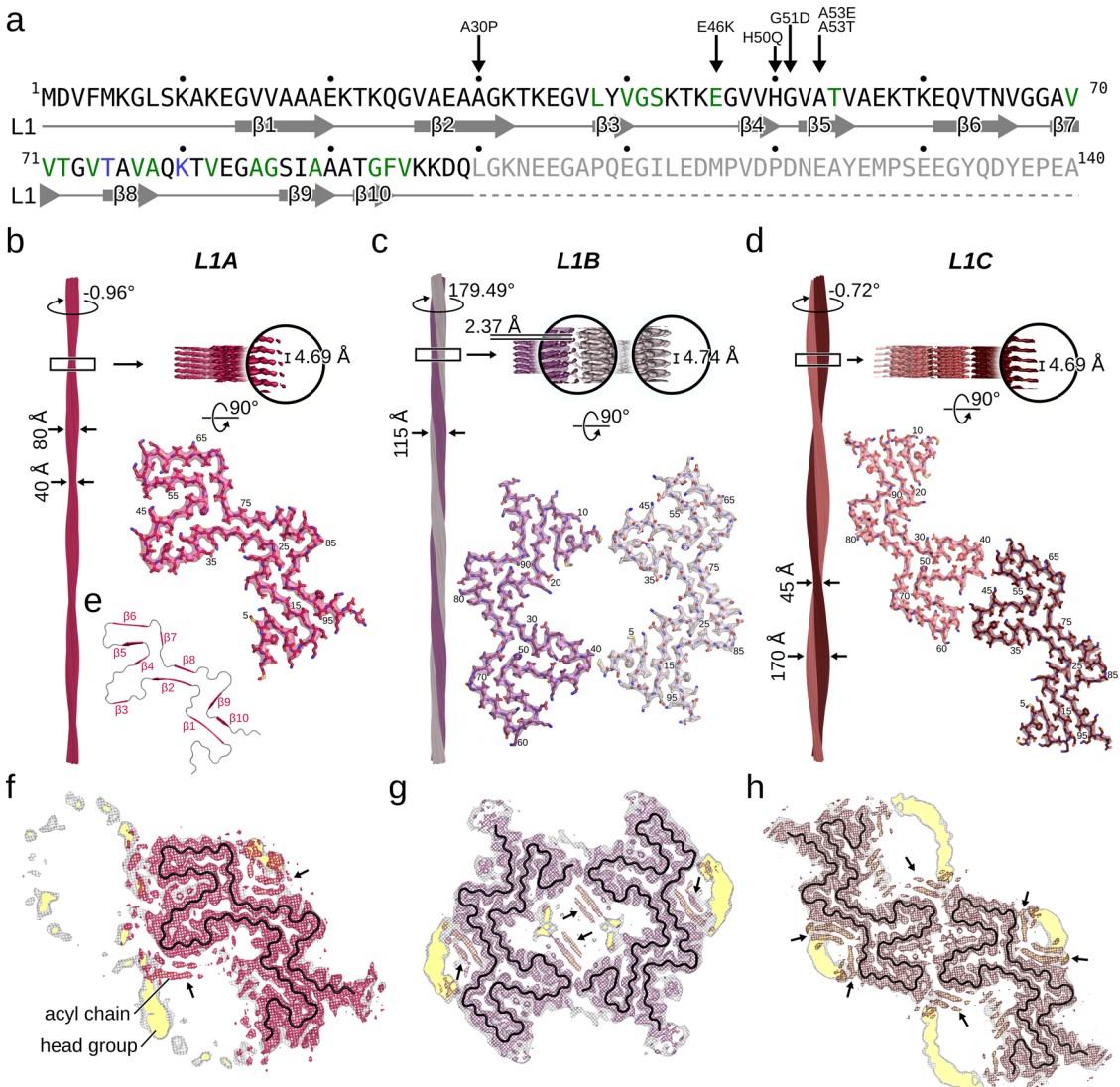

**Fig. 1 | Cryo-EM structures of the L1 αSyn fibrils. a** Sequence and secondary structure of human αSyn. Familial PD mutation sites (black arrow) localized within the lipid binding N-terminal region (residues 1–60)[64]. Green-colored residues bind to the lipid acyl chain, blue to the choline moiety[20], and gray were not resolved. **b–d** Cryo-EM structures of L1A (**b**), L1B (**c**), and L1C (**d**) fibrils (protofilaments colored differently). The atomic models are shown as sticks. Labels denote the fibril width, the helical twist and rise, and residue numbers. The density maps in the lower panels are displayed using the carve feature in PyMOL at a distance of 2 Å. **e** Backbone of the L1 fibrils with the β1–β10 colored magenta and loops in gray. **f–h** Overlay of a sharpened high-resolution map shown in magenta (**f**), purple (**g**), and brown (**h**) and an unsharpened, 4.5 Å low-pass filtered density in gray. The backbone is shown as a black ribbon. Densities highlighted with a yellow background are reminiscent of lipid micelles.

from MD simulations also reflects the periodical arrangement of the rod-shaped densities along the helical axis seen in the cryo-EM maps. Consequently, the MD data corroborates our assumption that the non-proteinaceous densities in the cryo-EM cross-sections likely are phospholipids bound to the fibril surface. In particular, the ring-shaped densities in the cryo-EM cross-sections likely constitute the lipid head groups, while the rod-shaped densities can be attributed to the lipid acyl chains.

The patterns of lipid interactions per residue repeated in all L1 fibrils suggest that lipid-mediated intramolecular interactions may be necessary for the yet unobserved L1 folding. For all L1 lipids, the predominantly hydrophobic segments $_1$MDVFM$_5$, $_{36}$GVLYV$_{40}$, $_{69}$AVVTGV-TAVA$_{78}$, and $_{85}$AGSIAAATGFV$_{95}$ are in contact with the lipid acyl chains. At the same time, the adjacent polar residues K6, E20, K21, K32, E35, N79, K80, and S87 interact with the lipid head groups (Fig. 4). Hence, hydrophobic areas on the fibril surface are, at least partially, covered with phospholipids. Figure 5 shows a POPC molecule modeled into the most well-defined non-proteinaceous densities at the fibril surface.

The central cavity in the L1B fibril is occupied by lipids, with their head groups bridging interactions between K6, K21, E20, and E35, while their acyl chains form hydrophobic interactions with M1, V2, M5, G36, L38, and V40 bridging across the protofilament interface (Figs. 3d, e, 4). The MD simulations revealed that the L1B cavity is occupied by chloride ions (Cl$^-$), which are complexed by the positively charged residues K21 and K23. For the L1C fibril, also revealed a high probability for Cl$^-$ ions in the hydrophilic interface involving residues K43, K45, and E57 is found (Figs. 3f, g, 4).

A striking feature of the L2A fibril is the bridging of lipid molecules that span the ~20 Å gap between the protofilaments. The simulations revealed that lipids interact with the segment $_{33}$TKEGVLYVGSKTK$_{45}$, bridging the gap between the protofilaments. In detail, the acyl chains bind to Y39, V40, and G41, which form a small hydrophobic patch at the fibril surface (Fig. 4). Additionally, the lipid head groups interact with K43 and K45 on one protofilament and with K34 on the neighboring protofilament (Fig. 4). The head group densities of these lipids partially overlap with densities for Cl$^-$ (Fig. 3h) and the per-residue

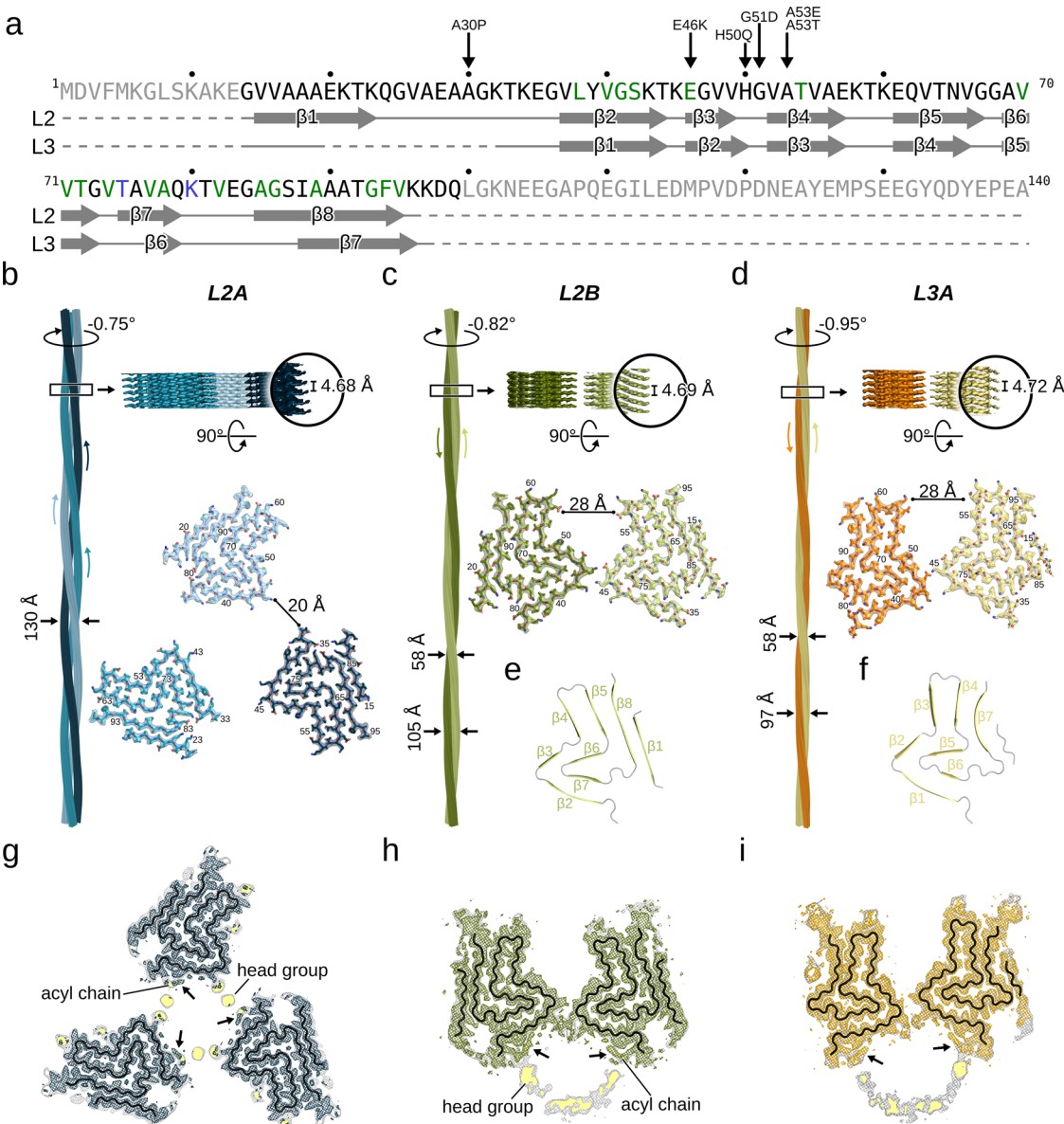

**Fig. 2 | Cryo-EM structures of L2 and L3 αSyn fibrils. a** See Fig. 1a for details. **b–d** Cryo-EM structures of L2A (**b**), L2B (**c**), and L3A (**d**) fibrils (protofilaments colored differently). The atomic models are shown as sticks. Labels denote the fibril width, the helical twist and rise, and residue numbers. The density maps in the lower panels are displayed using the carve feature in PyMOL at a distance of 2 Å. **e, f** Backbone trace of the L2 (**e**) and L3 (**f**) fibrils with the β1 - β8 colored green or yellow and loops in gray. **g–i** Overlay of a sharpened high-resolution map shown in blue (**g**), green (**h**), and orange (**i**) and an unsharpened, 4.5 Å low-pass filtered density is shown in gray. The backbone of the model is shown as black ribbon. Unsharpened densities highlighted with a yellow background are reminiscent of lipid micelles.

analysis confirmed that K34, K43, and K45 also interact with Cl⁻ (Fig. 4). Hence, the negatively charged phosphate groups and the Cl⁻ ions together form the bridge between K34 and K43 in the individual protofilaments by forming a well-ordered interaction network.

Although the L2 and L3 folds appear reminiscent of reported structures[26,27], lipid-fibril interactions favor alternative quaternary protofilament arrangements. In the L2A fibril, lipid-mediated interactions seem to be essential as they connect the neighboring protofilaments. Lipid-mediated interactions might also be responsible for the protofilaments pointing in opposite directions in the L2B and L3A fibrils, as in this configuration, two mirrored $_{34}$KEGVLYVGSK$_{43}$ segments from both protofilaments are in contact with the same phospholipid micelle (Fig. 3i, j). Again, the acyl chains bind to Y39, V40, and G41, the head groups interact with K34, K43, and K45 on both protofilaments, and Cl⁻ ions colocalize with head groups at the interface between K43 and K45 (Fig. 4).

While micelle-like lipid arrangements at the fibril surface can potentially also result from diffusion of lipid molecules after fibril formation, lipids in the central cavity of L2A seem less likely to originate from this process, as lipids mediate the interaction between the protofilaments, suggesting the presence of lipids already during fibril assembly. It is curious to note that the segment $_{35}$EGVLYV$_{40}$ in the lipidic L1 fibrils or $_{34}$KEGVLYVGSK$_{43}$ in lipidic L2 and L3 fibrils are in contact with lipids, which suggests that this stretch of residues could play an important role throughout fibril formation. Indeed, residues within this sequence, such as Y39, have previously been identified to play a crucial role in lipid binding, aggregation kinetics, and function[31,32].

Although the mechanisms of αSyn aggregation and toxicity in vivo are still under debate[33,34], disruption of intracellular vesicles is one potential mechanism for the cellular toxicity mediated by αSyn fibrils[19] as well as oligomers[18]. Reynolds et al. proposed a mechanism in which the abnormal aggregation of αSyn is linked to continuous lipid

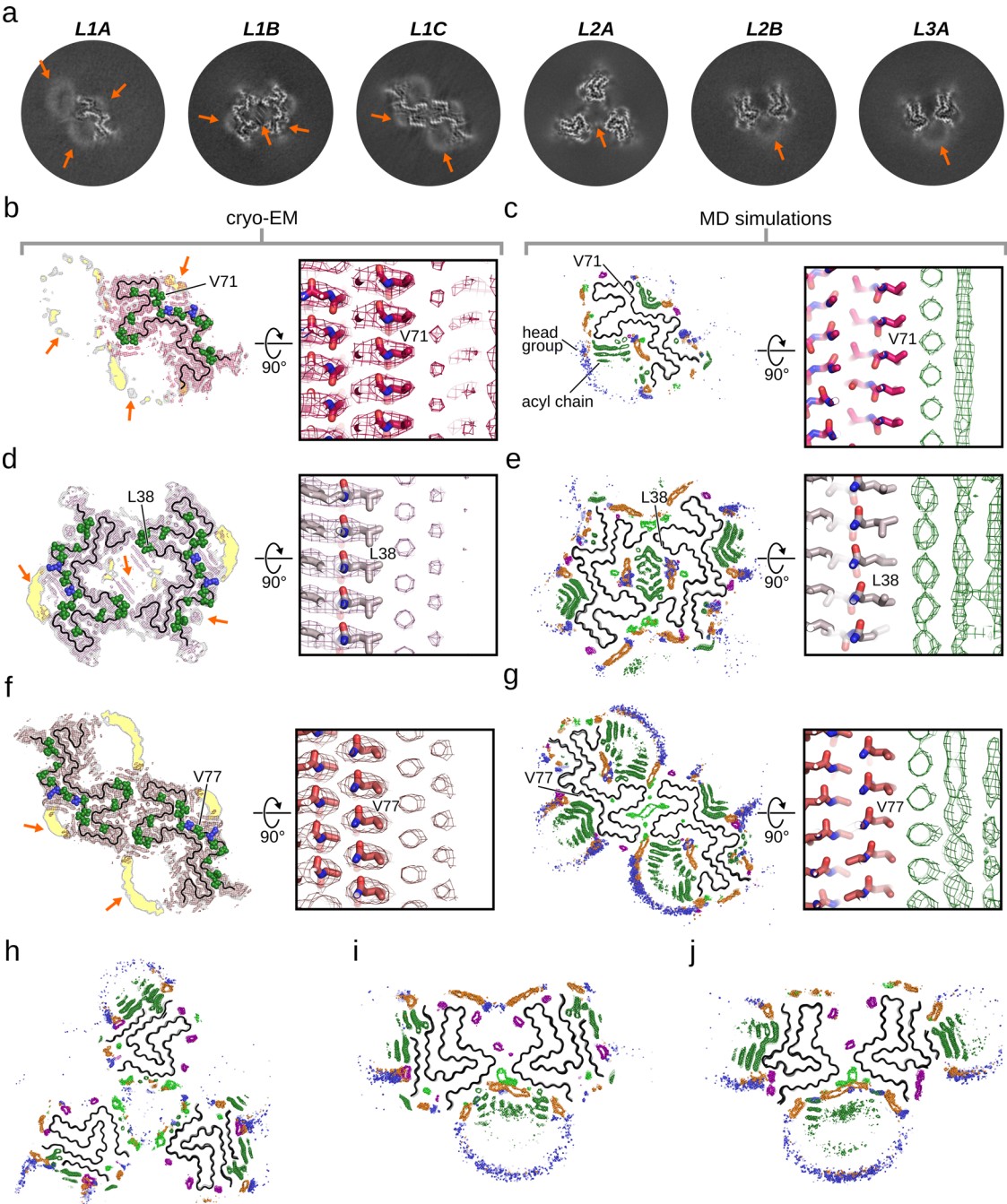

**Fig. 3 | Lipid-fibril interactions. a** Central slice of the unsharpened refined cryo-EM maps. **b**, **d**, **f** Superposition of the reconstructed cryo-EM maps and the atomic model. Sharpened, high-resolution maps are shown in magenta (**b**, L1A), violet (**d**, L1B), and red (**f**, L1C). Unsharpened, 4.5 Å low-pass filtered density is shown in gray. The backbone of the model is shown as black ribbon, with residues binding to the acyl chain (green) or choline moiety (blue) of phospholipids shown as spheres[20]. In (**a**, **b**, **d**, **f**) the arrows highlight non-fibrillar densities. **c**, **e**, **g** The grids indicate the probability density of the lipid acyl chain (dark green), phosphate (orange), and the choline nitrogen (blue), and sodium (purple), and chloride (light green) ions throughout MD simulations of the L1A (**c**), L1B, (**e**), and L1C (**g**) fibril. In (**b**–**g**) the right panels show a close-up view visualizing the ordered packing of the lipid molecules along the helical axis. **h**–**j** Probability density of the lipids throughout MD simulations of lipid diffusion for the L2A (**h**), L2B, (**i**), and L3A (**j**) fibril.

extraction mediated by the growing aggregates, eventually leading to membrane disruption[35]. The finding of direct lipid-fibril interactions due to their lipid-associated aggregation may provide the structural basis for the proposed lipid extraction mechanism[35]. In addition, the lipid-coated fibrils reported here give a structural rationale to the previously suggested lipid co-aggregation with αSyn fibrils[36].

In recent years, a growing number of ex vivo cryo-EM fibril structures have been discovered that are characteristic of different diseases[37–39]. That none of these so far are lipidic fibrils might be explained by the use of detergent during the isolation of fibrils from patient tissue.

In conclusion, we report six cryo-EM structures of lipidic αSyn fibrils, revealing how lipid molecules bind directly to the fibril surface. Insights obtained from these lipidic fibrils emphasize that studying αSyn aggregates in the presence of lipids is relevant for understanding the molecular basis of α-synucleinopathies. Furthermore, modulation of lipid-fibril interactions may also provide a promising strategy in searching for therapeutic interventions.

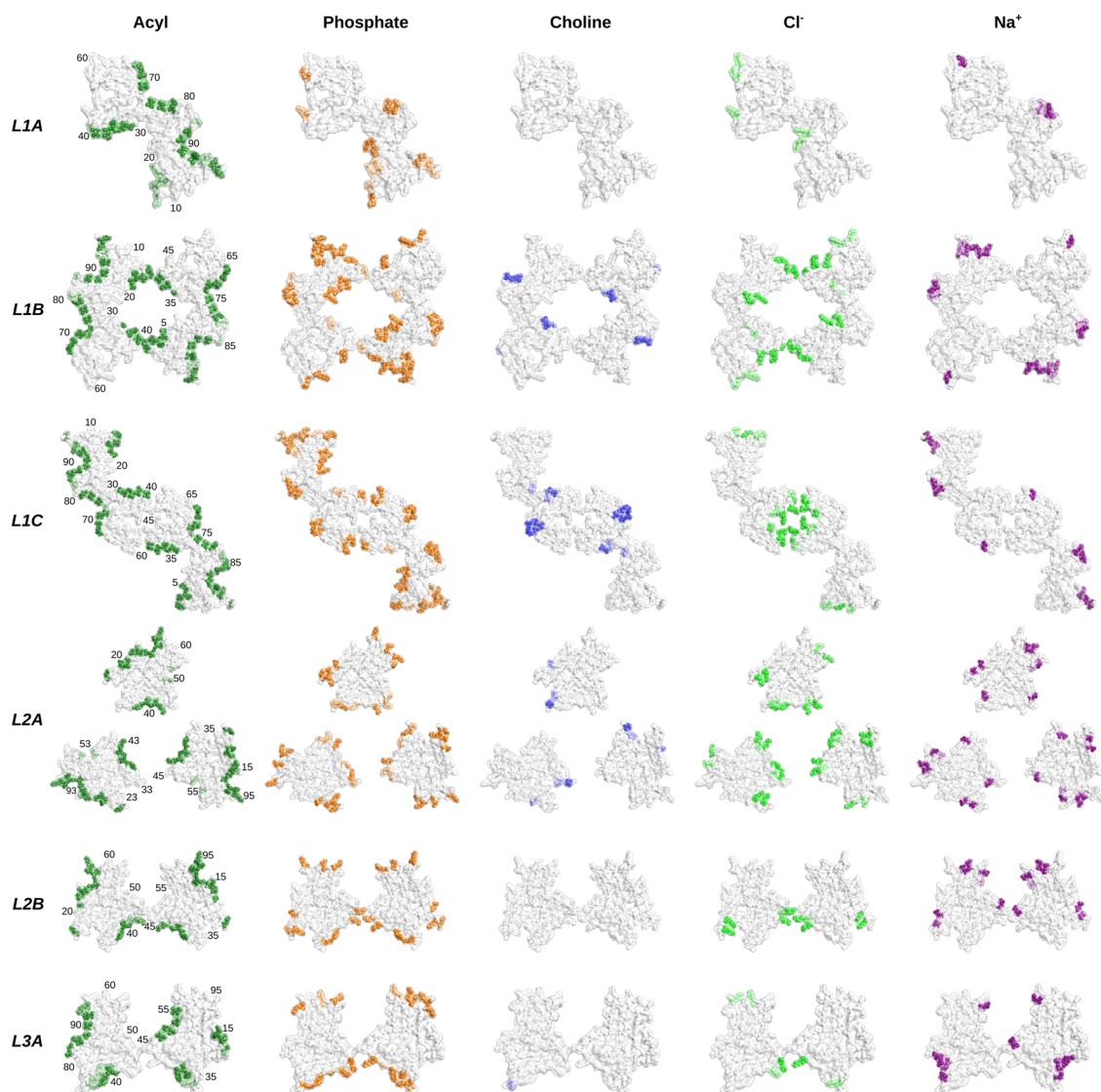

**Fig. 4 | Per-residue interactions with phospholipids and ions.** Atomic models with residues colored according to their interaction frequencies with the acyl chains (green), phosphate group (orange), quaternary choline group (blue), chloride (light green), and sodiums ions (purple) throughout the MD simulations. Residues that interact in at least 50% of all conformations are colored with the color saturation linearly increasing with interaction frequencies between 50 and 100%. The remaining residues are colored white.

## Methods

### Protein expression and purification

αSyn was expressed recombinantly in E. coli strain BL21(DE3) and purified as described previously[40]. Briefly, the protein was expressed in minimal medium at 37 °C. Cells were harvested 6 h after induction, lysed by freeze-thaw cycles followed by sonication, boiled for 15 min and centrifuged at $48,000 \times g$ for 45 min. From the supernatant DNA was precipitated with streptomycin (10 mg/ml) while stirring the ice-cold solution. After centrifugation αSyn was precipitated from the supernatant by adding ammonium sulfate to 0.36 g/ml. After another centrifugation step the pellet was resuspended in 25 mM Tris/HCl, pH 7.7 and the protein was further purified by anion exchange chromatography on a 30 ml POROS HQ column (PerSeptive Biosystems). To prepare monomeric αSyn without any aggregates, the protein was dialyzed against PBS buffer, pH 7.4, centrifuged at $106,000 \times g$ for 1 h at 4 °C and filtrated through 0.22 μm ULTRAFREE-MC centrifugal filter units (Merck Millipore). The final protein concentration was adjusted to 0.33 mM.

### Preparation of αSyn fibrils

Samples of αSyn fibrils were prepared as previously reported[20]. In brief, vesicles were prepared by mixing 1-palmitoyl-2-oleoyl-sn-gly-cero-3-phosphocholine (POPC), 1-palmitoyl-2-oleoyl-sn-glycero-3-phosphate (POPA, sodium salt) dissolved in chloroform respectively and evaporating the solvent under a $N_2$-stream followed by lyophilization overnight. SUVs were obtained by repeated sonication of a solution of 1.5 mM POPC, 1.5 mM POPA. Vesicles were incubated with 70 μM $^{13}C$, $^{15}N$-labeled αSyn in buffer (50 mM HEPES, 100 mM NaCl, pH 7.4) at a lipid to protein ratio of 5:1 and subjected to repeated cycles of 30 s sonication (20 kHz) at 37 °C followed by an incubation period of 30 min. After 24 h (dataset 1), 48 h (dataset 2) and 20 h (dataset 3) respectively the samples were transferred to a Multitron incubator (Infors HT, Bottmingen, CH) and shaken at 100 rpm (50 mm throw) at 37 °C until a combined aggregation time of 96 h was reached. Aggregation was monitored regularly by mixing 5 μL of the aggregate solution with 2 mL of Thioflavin T containing buffer (100 μM ThT, 50 mM Glycine, pH 8.5) and measuring the fluorescence emission intensity at 482 nm in a Varian Cary Eclipse fluorescence spectrometer.

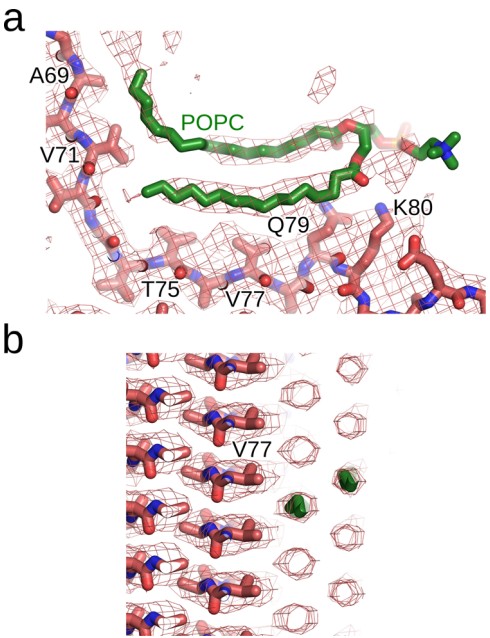

**Fig. 5 | A lipid-fibril binding-mode model.** One POPC molecule (green sticks) was modeled into the sharpened map of L1C and is shown in a view along the fibril axis (**a**) and perpendicular to the fibril axis (**b**).

For cryo-EM samples 700 μL of aggregate solution were then centrifuged for 5 min at $16,873 \times g$ in a F-45-18-11 Rotor in a 5418 R tabletop centrifuge (Eppendorf, Hamburg, GER). If fibrils did not pellet right away, the procedure was repeated until a visible pellet was obtained. The supernatant was removed and 50 μL of fresh buffer (5 mM HEPES, pH 7.4) were added and thoroughly mixed with the pellet to obtain a highly concentrated fibril solution.

For ssNMR samples a minimum of 1500 μL of the aggregate solution were centrifuged at $152,460 \times g$ (TLA-100.3 rotor in an Optima™ MAX-TL) for 1 h at 4 °C. After removal of the supernatant, samples were washed with fresh buffer (5 mM HEPES, pH 7.4) and subsequently centrifuged (10 min, $212,940 \times g$, 18 °C). Excess moisture was carefully removed, and samples were packed into either 1.3 mm or 3.2 mm ssNMR rotors by cutting off the bottom of the tube and centrifuging the pellet directly into the rotor of choice through a custom-made filling device made from a truncated pipette tip. Finally, the sample was centrifuged into the rotor in an ultracentrifuge packing device for 30 min at $98,381 \times g$ in a SW 32 Ti rotor in an Optima™ L-80 XP Ultracentrifuge (both Beckman Coulter)[41].

### ssNMR
3D (H)CANH experiments[42] $^{13}$C, $^{15}$N-labeled αS on an 800 MHz Bruker Avance III HD spectrometer at a magnetic field of 18.8 T or a 1200 MHz Bruker Avance NEO spectrometer at a magnetic field of 28.2 T each equipped with a 1.3 mm magic-angle spinning (MAS) HCN probe and MAS at 55 kHz. The temperature of the cooling gas was set to 250 K, resulting in an estimated sample temperature of 20 °C.

2D (H)NCA spectra were acquired on an 850 MHz Avance III spectrometer with a 3.2 mm MAS HCN probe at a magnetic field of 20.0 T and MAS at 17 kHz. The temperature of the cooling gas was set to 265 K, resulting in an estimated sample temperature of 20 °C.

$^{1}$H decoupled $^{31}$P spectra were acquired on an 600 MHz Avance III spectrometer with a 1.3 mm MAS HCN probe (equipped with a range coil for $^{31}$P tuning) at a magnetic field of 14.1 T without MAS. The temperature of the cooling gas was set to 278.2 K and 310.2 K, resulting in estimated sample temperatures of 7 °C and 37 °C respectively. For spectra of vesicles, SUVs were prepared as described above. The resulting solution was lyophilized and resuspended in drops buffer

(10 mM HEPES) to increase concentration. The resulting gel was centrifuged into the rotor in an ultracentrifuge packing device as described above.

### Cryo-EM grid preparation and imaging
For cryo-EM grid preparation, 1.5 μL of fibril solution were applied to freshly glow-discharged R2/1 holey carbon film grids (Quantifoil). After the grids were blotted for 12 s at a blot force of 10, the grids were flash frozen in liquid ethane using a Mark IV Vitrobot (Thermo Fisher) operated at 4 °C and 95% rH.

Cryo-EM datasets were collected on a Titan Krios transmission-electron microscope (Thermo Fisher) operated at 300 keV accelerating voltage and a nominal magnification of 81,000× using a K3 direct electron detector (Gatan) in non-superresolution counting mode, corresponding to a calibrated pixel size of 1.05 Å. Data was collected in EFTEM mode using a Quantum LS energy filter at a slit width of 20 eV. A total of 11,740, 7836, and 7744 movies were collected with SerialEM[43] for Datasets 01, 02 and 03, respectively. Movies of Dataset 01 were recorded over 50 frames accumulating a total dose of ~51 e⁻/A², whereas movies of Dataset 02 and 03 contained 40 frames with a total dose of ~43 e⁻/A². The range of defocus values collected spans from −0.5 to −2.0 μm. Collected movies were motion corrected and dose weighted on-the-fly using Warp[44].

### Helical reconstruction of αSyn fibrils
αSyn fibrils were reconstructed using RELION-3.1[45], following the helical reconstruction scheme[46]. Firstly, estimation of contrast transfer function parameters for each motion-corrected micrograph was performed using CTFFIND4[47]. For filament picking, we only considered micrographs with an estimated resolution of ≤3.8 Å (Dataset 01), ≤4.0 Å (Dataset 02), and ≤5.0 Å (Dataset 03), respectively (Supplementary Table 1).

For 2D classification, we extracted particle segments using a box size of 600 pix (1.05 Å/pix) downscaled to 200 pix (3.15 Å/pix) and an inter-box distance of 13 pix. L1A, L1B, L1C, L2A fibrils were successfully separated at this 2D classification stage, whereas L2B and L3A were too similar on the 2D level.

For 3D classification, the classified segments after 2D classification were (re-)extracted using a box size of 250 pix (1.05 Å/pix) and without downscaling. Starting from featureless cylinder filtered to 60 Å, several rounds of refinements were performed while progressively increasing the reference model's resolution. The helical rise was initially set to 4.75 Å and the twist was estimated from the micrographs. Once the β-strands were separated along the helical axis, we optimized the helical parameters (final parameters are reported in Supplementary Table 1). During 3D classification, we successfully separated L2B and L3A fibrils, which were then treated individually. We performed multiple rounds of 3D auto-refinement from here on until no further improvement of the map was observed. Standard RELION post-processing with a soft-edged solvent mask that includes the central 10% of the box height yielded post-processed maps (B-factors are reported in Supplementary Table 1). The resolution was estimated from the value of the FSC curve for two independently refined half-maps at 0.143 (Supplementary Fig. 3). The optimized helical geometry was then applied to the post-processed maps yielding the final maps used for model building. For all fibrils we found regions in the structure where the local resolution is sufficient to identify density for the backbone carbonyl groups, and with this all fibrils were found to have a left-handed twist.

### Determination of the relative population of each fibril polymorph
In the cases of L1A, L1B, L1C, and L2A fibrils, the population relative to the total number of extracted helical segments was calculated based on the number of helical segments after the successful separation by 2D classification. As to L2B and L3A, on the other hand, we used the

number of helical segments after successful separation by 3D classification.

## Atomic model building and refinement

The atomic models of L1 fibrils were built de novo in Coot[48]. For L2 fibrils, one protein chain was extracted from PDB ID 6SST[26] of wild type αSyn and used as the initial model. For L3 fibrils, one protein chain from PDB ID 6UFR[27] of E46K αSyn was extracted and used as the initial model. To the latter, the amino acid sequence was converted to wild type αSyn (UniProt: P37840) and the N-terminal region G14 to A19 was built de novo in Coot[48]. Subsequent refinement in real space was conducted using PHENIX[49,50] and Coot[48] in an iterative manner. The resulting models were validated with MolProbity[51] and details about the atomic models are described in Supplementary Table 2.

To visualize the lipid interactions, we used the sharpened L1C map and initially modeled a POPC molecule into the density, again using Coot[48]. Subsequently, another round of real space refinement was conducted using PHENIX[49,50].

## Molecular dynamics simulations of lipid diffusion

To investigate where and how the lipids interact with the different types of αSyn fibrils, we performed unbiased molecular dynamics (MD) simulations of POPC and POPA in the presence of the αSyn fibrils. A filament was always composed of 20 helically arranged peptide chains. Except for residue M1 in L1 fibrils, ACE- and NME-caps were connected to the N- and C-termini, respectively, to avoid artificially charged termini.

We then used PACKMOL[52] to, first, center the αSyn fibril in a rectangular simulation box, and, second, to randomly place POPC and POPA lipids, sodium ($Na^+$) and chloride ($Cl^-$) ions, and water molecules around the αSyn fibril. We added additional $Na^+$ or $Cl^-$ counter ions to enforce the neutrality of the systems. In the final setup, we mimicked the experimental conditions used for αSyn fibril aggregation[20], meaning that side chains are prepared for pH 7.4, the NaCl concentration is 100 mM, and a molar lipid/protein ratio is 10 (ratio of 1:1 for the lipids).

The Amber ff19SB force field[53] was applied to describe the αSyn fibrils and the Lipid17 force field[54] to describe the POPC and POPA molecules. Ion Parameters for monovalent ions were taken from ref. 55 and used in with the OPC water model[56].

The exact minimization, thermalization (towards 300 K), and density adaptation (towards 1 g/cm$^3$) protocol is reported in ref. 57, which was applied previously to study ligand binding processes to amyloid fibrils[29] (Supplementary Fig. 9). The conformations after thermalization and density adaptation served as starting points for subsequent NPT production simulations. Therefore, the initial velocities were randomly assigned during the first step of the following NPT production simulation, such that each simulation can be considered as an independent replica. For each αSyn fibril, we completed eight independent NPT production simulations at 300 K and 1 bar for 1 μs each. Importantly, we restrained the backbone to the initial atomic coordinates, as the fibril models used for MD simulations were not stable without the final proper arrangement of lipids around the fibrils, which were not known at the beginning of the simulations. However, all other molecules, including POPC and POPA, were allowed to diffuse freely and we did not apply any artificial guiding force. During production simulations, Newton's equations of motion were integrated in 4 fs intervals, applying the hydrogen mass repartitioning approach[58] to all non-water molecules, which were handled by the SHAKE algorithm[59]. Coordinates were stored into a trajectory file every 200 ps. The minimization, thermalization, and density adaptation were performed using the pmemd.MPI[60] module from Amber20/AmberTools21[61], while the production simulations were performed with the pmemd.CUDA module[62]. To further test the fibril stability, we performed additional simulations without lipids and without positions

restraints and found that the quaternary arrangement of the fibrils is not stable in that case (Supplementary Fig. 10).

## Determination of the binding region for lipids

We used cpptraj[63] from Amber20/AmberTools21[61] to calculate 3D density grids (normalized to the number of considered conformations) separately for the lipids' acyl chain, the phosphate atom, and the choline nitrogen atom. These grids represent the probability density of a molecule position relative to the centered fibril structure. Initially, we calculated the 3D density grids for each trajectory, constantly increasing the time range for the analysis in 0.1 μs intervals. Thereby, we observed only minimal changes when extending the analysis time from 0.9 μs to 1.0 μs, such that we assumed converged distributions of the lipid molecules (Supplementary Fig. 11). Hence, the average density grids were calculated over all conformations of the 0.9 to 1.0 μs interval of all MD simulations replicates.

Finally, we calculated the average interaction frequencies for every amino acid with POPC, POPA, $Na^+$, and $Cl^-$. For this, we measured the minimum distance between any non-hydrogen atom of every amino acid of five layers from the center of each protofilament to (i) the phosphate group of the phospholipids, (ii) the quaternary choline group of the phospholipids, (iii) any carbon atom of the acyl chains of the phospholipids, (iv) any $Na^+$, and (v) any $Cl^-$ ion. An interaction was present, if the distance was smaller than 5 Å. These interactions are normalized by the total number of frames, so that a value of 1.0 means "interaction always present", whereas a value of 0.0 means "interaction not existent". We considered an amino acid as "interacting", if the interaction is present in at least 50% (value 0.5) of all conformations and "strongly interacting" if the interaction is present in at least 75% (value 0.75) of all conformations. Again, our analysis focusses on the 0.9 to 1.0 μs interval of all MD simulations replicates.

## Reporting summary

Further information on research design is available in the Nature Portfolio Reporting Summary linked to this article.

## Data availability

The cryo-EM maps have been deposited in the Electron Microscopy Data bank (EMDB) under the accession numbers EMD-15370 (L1A), EMD-15371 (L1B), EMD-15372 (L1C), EMD-15148 (L2A), EMD-15369 (L2B), and EMD-15388 (L3A). The corresponding atomic models have been deposited in the Protein Data Bank (PDB) under the accession numbers: 8ADU (L1A), 8ADV (L1B), 8ADW (L1C), 8A4L (L2A), 8ADS (L2B), and 8AEX (L3A). NMR Spectra raw data generated in this study have been deposited in the open research data repository Edmond at https://doi.org/10.17617/3.9YH1RW. Supplementary Information is available for this paper, including a Supplementary Movie S1 and the Supplementary Legend to Supplementary Movie S1. Source data are provided with this paper.

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

## Acknowledgements
This work was supported by the Max Planck Society (to CG) and the Deutsche Forschungsgemeinschaft (DFG, German Research Foundation) under Germany's Excellence Strategy-EXC 2067/1-390729940 (to CG) and the Emmy Noether program to LBA (project number: 397022504). BF, JAG, and GFS are grateful for computational support and infrastructure provided by the "Zentrum für Informations- und Medientechnologie" (ZIM) at the Heinrich Heine University Düsseldorf and the computing time provided by Forschungszentrum Jülich on the supercomputer JURECA/JURECA-DC at Jülich Supercomputing Center (JSC). We thank Karin Giller and Melanie Wegstroth for excellent technical help with protein sample preparation.

## Author contributions
C.G. and G.F.S. designed and supervised the project. B.F., Le.A., C.G., and G.F.S. administered the project. S.B. performed protein expression and purification. Le.A. prepared the fibril samples. Le.A., E.E.N., and L.B.A. recorded the ssNMR data and analyzed the associated data. C.D. prepared the cryo-EM grids and collected the cryo-EM images. B.F. processed the cryo-EM images, reconstructed the fibril structures, built the atomic models, performed the MD simulations, and analyzed the associated data. B.F. and Le.A. visualized the results. B.F., Le.A., C.G., and G.F.S. wrote the original draft. All authors reviewed and edited the paper.

## Funding

## Competing interests
The authors declare no competing interests.
