## [Peer Review File · Nature Communications]

REVIEWER COMMENTS

Reviewer #1 (Remarks to the Author):

Frieg et al. present cryo-EM analysis of phospholipid-induced α -synuclein (lipid- α -syn) fibril structures, and reveal the structural basis of the interaction between lipid and α -syn in fibrillar form from the additional cryo-EM densities of on lipid- α -syn fibril together with MD simulations and ssNMR data. The authors determine six lipid- α -syn fibril complex structures. Interestingly, three L1 fibrils possess a novel protofilament fold type, which expands the structural knowledge on the structural polymorphism of α -syn amyloid fibril. L2 and L3 feature previously known α -syn fold in fibril structures, but with distinct protofilamental arrangement. Moreover, the MD simulations of lipid diffusion in the presence of α -syn fibril structure suggest a micelle-like lipid arrangements at the fibril surface and the central cavity, which matches the cryo-EM densities well. By further performing ssNMR, the authors assign the fibril-lipid interactions for each fibril polymorph. Both lipid and α -syn fibrils are enriched in Lewy bodies in the patients' brains of Parkinson's disease. Elucidating lipid- α -syn fibril interaction is important to understanding the molecular basis of α -syn pathological aggregation and Lewy body formation in PD. Thus, this work is timely and important to the field. Overall, the results are well presented in a logic format, and the complex structural models are cross-validated by different biophysical and computational methods. To strengthen this work, the authors may need to address my concerns listed below.

Major concerns:

1. The authors perform MD simulations for phospholipid diffusion and show probability densities of lipid, acyl chain, phosphate, choline nitrogen, chloride, and sodium. The results match the non-proteinaceous densities of cryo-EM maps in both cross-section view and axial view. To further confirm that the extra densities are from lipids, the authors modeled POPC/POPA molecule into the well-defined densities of each polymorph, as in Fig. 4, which may indicate the conformation of POPC/POPA molecule. Moreover, the detailed structural analysis about the interaction between α -syn fibrils and the docked POPC/POPA is absent. Additional interaction analysis may provide a clear view of how the micelle-like lipids pack on the fibril surface.
2. It seems that the structural model coordinates and the cryo-EM maps have not been submitted to PDB or EMDB. I strongly suggest the authors upload the files to these two databases and provide full wwPDB validation reports.
3. Whether POPC/POPA vesicles affect the fibrillation kinetics of α -syn? The authors might need to perform ThT fluorescence assay to monitor the α -syn fibrillation kinetics in the presence of lipids.
4. The authors might need to compare the L1 fibril fold with previously reported different α -syn folds, and discuss what's the novel structural feature of this fold.
5. The distinct packing pattern of protofilaments in L2B and L3A fibrils is interesting and features a novel protofilamental packing symmetry. How did the authors determine the handedness of these fibrils? Atomic force microscope can be used to characterize the handedness of fibril twist. For high resolution maps, densities for the carbonyl oxygen atoms may also help to confirm the handedness.
6. The authors need to clarify how the relative population of each lipid- α -syn fibril polymorphs (Extended Data Fig. 1g) is determined.
7. As for the statement "In the L1B fibril, both protofilaments are related by an approximate 21 screw symmetry and the protofilaments are tilted by $\sim 37^\circ$ to each other.", how is the tilt angle measured?

Minor concerns:

1. As for L1C, "ionic interactions between residues K45 and E46 form the inter-protofilament interface" should be "K43, K45 and E57";
2. Twist angle of L1A in Figure 1b should be consistent with the number in Extended Data Table 4;
3. Initial model used for model building of L3A should be 6UFR;
4. The tables cited in the main text is not in a right order.

Reviewer #2 (Remarks to the Author):

The manuscript describes cryo-EM structures of six α -synuclein fibril polymorphs in complex with lipids. The phospholipids are suggested to induce (some) new morphologies of the fibrils and to bind to cavities within them. The authors suggest that the structures support a mechanism of co-aggregation of lipids with fibril, and to fibril-induced lipid extraction, leading to cell toxicity and pathology via disruption of intracellular vesicles.

The "big question" here is whether the lipids really induce specific conformations that are relevant to membrane interactions or rather are just "additional polymorphs" of α -syn that are induced by different conditions or a part of an inherent population mix. I think it is impossible to answer this question with current tools. There are few residues that were correlated with lipid interactions, but this is a very limited information and support. The most convincing evidence is the presence of lipids in the central cavity of L2A that might mediate the interaction between the protofilaments. Having said that, the L2A protofilament arrangement was observed without lipids, then yet the intertwining of the protofilaments might be lipid induced and relevant. Really impossible to tell. I agree that it is tempting to suggest that the structures support a proposed lipid extraction mechanism and co-aggregation of lipids with fibrils.

1. Another conceptual question is the colloidal state of the lipids which encounter the protein/fibrils. Is it in the presence of vesicles or micelles (probably affected by the sonication/incubation)? Or some other type of colloidal system? This might be relevant in case the interaction / state of the lipids after sonication is different in comparison to a vesicular system which resembles more physiological condition. Why didn't you use intact vesicles?

Technical comments:

2. Fig 3 - "In a+d, the arrows highlight non-fibrillar densities" - should it be a-f? I think there is a mix-up in the colours of the map in the figure legend.
3. The residues that were identified by ssNMR to bind to the PLs are not named explicitly except from within the image itself (3 hydrophobic residues? - what about the headgroups?).
4. Fig 3d,e - points to L1B and it is refer in the text discussing K6, K21, and E20 - but these are not shown.
5. Fig 3f - L1C fibril - K43, K45, and E57 should be shown in closeness to the Cl ion as discussed in the text.
6. Fig 3h - L2A - it is impossible to appreciate the real distance between the lysines and the Cl ions in this figure. "where Cl- ions colocalize with head groups at the interface between K43 and K45 (Fig. 3a,i,j)" - one cannot see it clearly in 3i,j and definitely not in 3a. zoom-in is needed.
7. Segments 35-EGVLYV-40 and 34-KEGVLYVGSK-45 should be indicated in the figures. Also a zoom-in of Y39.
8. Please indicate the FSC=0.143 resolution in Extended Data Fig. 2 (crossing lines or something similar).
9. I am confused by the 2.37Å helical rise of L1B. In the FSC there is a peak at ~4.7Å which is similar to the other polymorphs.
10. In page 6 line 4 is stated: "The conversion of the SUVs used for the preparation of the lipidic fibrils to such small lipid aggregates upon fibril formation was confirmed by 31P ssNMR", so it means that the SUVs were used as initial system? Is it the same lipid state as used for the cryo-EM?
11. Why are the PDB reports "*Not For Manuscript Review? I have never seen this statement. There are many clashes, but I think mostly of hydrogens, so it is fine. Anyway, the resolution seems to be good enough to avoid clashes.

Reviewer #3 (Remarks to the Author):

The authors of the study present 6 new cryo-electron microscopy α -synuclein fibril structures. They also support their experimental work with MD simulations. The study is well-structured, up-to-date and original, and provides new insights on lipid- α -synuclein fibril interactions that could guide future therapeutic applications for PD. However, authors should address the following

questions before work is accepted

- Although the authors gave a reference, how many ns did it reach thermalization and density adaptation in MD simulations? this time period should be included in the manuscript. Also, is it possible to embed a visual proof of these metrics on the supporting information?
- In which statistical ensemble did the authors perform their equilibration simulations? this must be stated in the manuscript.
- It was interesting that the MD replicas were produced at the start of the NPT production simulations. Could the replicas have been produced at the start of the balancing simulations? Can the authors review this situation? Also, how were the initial velocities for these replicas randomly determined? Were they produced based on any physical function?
- The authors say that after 100 ns, the simulations convergence. Can they justify this situation with various visual metrics and put them in supporting information?
- In the simulation video, it looks like position constraints have been applied to the fibrils. If so, the authors should definitely state this situation in the manuscript along with their reasons.

Reviewer #4 (Remarks to the Author):

The authors present structures of alpha-synuclein fibrils prepared from in vitro incubation with phospholipids. This is highly significant work because it presents the first direct structural evidence of lipid interactions with synuclein fibrils and the structures are good quality.

Some drawbacks of the study are that the lipid to protein ratios are unusually low and there are large differences among the various structures observed, so it is not clear whether these actually represent situations observed physiologically. This criticism, however, is true for most alpha-synuclein structural studies, so it is not a fatal flaw.

The structures are well described and complementary MD simulations and solid state NMR studies support the overall impact of the work.

The validation reports indicate that they are not intended for submission to manuscripts so this seems awkward.

RESPONSE TO REVIEWERS' COMMENTS

We thank the four reviewers for the evaluation of our manuscript and for their many helpful comments and suggestions for improvements. In the following, the reviewer comments are in *italics* and our responses start with “**A:**”. Although not all text changes are reproduced in this response letter, selected parts of the revised manuscript text are highlighted in green. We believe that, thanks to the reviewer input, the manuscript has greatly improved.

Reviewer #1 (Remarks to the Author):

Frieg et al. present cryo-EM analysis of phospholipid-induced α -synuclein (lipid- α -syn) fibril structures, and reveal the structural basis of the interaction between lipid and α -syn in fibrillar form from the additional cryo-EM densities of on lipid- α -syn fibril together with MD simulations and ssNMR data. The authors determine six lipid- α -syn fibril complex structures. Interestingly, three L1 fibrils possess a novel protofilament fold type, which expands the structural knowledge on the structural polymorphism of α -syn amyloid fibril. L2 and L3 feature previously known α -syn fold in fibril structures, but with distinct protofilamental arrangement. Moreover, the MD simulations of lipid diffusion in the presence of α -syn fibril structure suggest a micelle-like lipid arrangements at the fibril surface and the central cavity, which matches the cryo-EM densities well. By further performing ssNMR, the authors assign the fibril-lipid interactions for each fibril polymorph. Both lipid and α -syn fibrils are enriched in Lewy bodies in the patients' brains of Parkinson's disease. Elucidating lipid- α -syn fibril interaction is important to understanding the molecular basis of α -syn pathological aggregation and Lewy body formation in PD. Thus, this work is timely and important to the field. Overall, the results are well presented in a logic format, and the complex structural models are cross-validated by different biophysical and computational methods. To strengthen this work, the authors may need to address my concerns listed below.

Major concerns:

1. The authors perform MD simulations for phospholipid diffusion and show probability densities of lipid, acyl chain, phosphate, choline nitrogen, chloride, and sodium. The results match the non-proteinaceous densities of cryo-EM maps in both cross-section view and axial view. To further confirm that the extra densities are from lipids, the authors modeled POPC/POPA molecule into the well-defined densities of each polymorph, as in Fig. 4, which may indicate the conformation of POPC/POPA molecule. Moreover, the detailed structural analysis about the interaction between α -syn fibrils and the docked POPC/POPA is absent. Additional interaction analysis may provide a clear view of how the micelle-like lipids pack on the fibril surface.

A: Following the reviewer's suggestion for a detailed structural analysis of α Syn-POPA/POPC interactions, we analyzed the MD trajectories towards residue-wise interactions not only with POPA and POPC, but also including Na^+ and Cl^- atoms. To do so, we measured the minimal distance between any non-hydrogen atom of every amino acid of five layers from the center of each protofilament to (i) the phosphate group of the phospholipids, (ii) the quaternary choline group

of the phospholipids, (iii) any carbon atom of the acyl chains of the phospholipids, (iv) any Na⁺, and (v) any Cl⁻ ion. An interaction was present, if the distance was smaller than 5 Å. These interactions are normalized by the total number of frames, so that a value of 1.0 means “interaction always present”, whereas a value of 0.0 means “interaction not existent”. We considered an amino acid as “interacting”, if the interaction is present in at least 50% (value 0.5) of all conformations and “strongly interacting” if the interaction is present in at least 75% (value 0.75) of all conformations. The results are shown in the new Fig. 4 (p. 15).

In addition, we extended the description and interpretation of the MD simulation data in the revised manuscript, which reads (pp. 6): “The patterns of lipid interactions per residue repeated in all *LI* fibrils suggest that lipid-mediated intramolecular interactions may be necessary for the novel *LI* folding. For all *LI* lipids, the predominantly hydrophobic segments ₁MDVFM₅, ₃₆GVLYV₄₀, ₆₉AVVTGVTAVA₇₈, and ₈₅AGSIAAATGFV₉₅ are in contact with the lipidic acyl chains. At the same time, the adjacent polar residues K6, E20, K21, K32, E35, N79, K80, and S87 interact with the lipidic head groups (Fig. 4). Hence, hydrophobic areas on the fibril surface are, at least partially, covered with phospholipids. Fig. 5 shows a POPC molecule modeled into the most well-defined non-proteinaceous densities at the fibril surface.

The central cavity in the *L1B* fibril is occupied by lipids, with their head groups bridging interactions between K6, K21, E20, and E35, while their acyl chains form hydrophobic interactions with M1, V2, M5, G36, L38, and V40 bridging across the protofilament interface (Fig. 3d, e, Fig. 4). The MD simulations revealed that the *L1B* cavity is occupied by chloride ions (Cl⁻), which are complexed by the positively charged residues K21 and K23. For the *L1C* fibril, also revealed a high probability for Cl⁻ ions in the hydrophilic interface involving residues K43, K45, and E57 is found (Fig. 3f,g, Fig. 4).

A striking feature of the *L2A* fibril is the bridging of lipid molecules that span the ~20 Å gap between the protofilaments. The simulations revealed that lipids interact with the segment ₃₃TKEGVLYVGSKTK₄₅, bridging the gap between the protofilaments. In detail, the acyl chains bind to Y39, V40, and G41, which form a small hydrophobic patch at the fibril surface (Fig. 4). Additionally, the lipidic head groups interact with K43 and K45 on one protofilament and with K34 on the neighboring protofilament (Fig. 4). The head group densities of these lipids partially overlap with densities for Cl⁻ (Fig. 3h) and the per-residue analysis confirmed that K34, K43, and K45 also interact with Cl⁻ (Fig. 4). Hence, the negatively charged phosphate groups and the Cl⁻ ions together form the bridge between K34 and K43 in the individual protofilaments by forming a well-ordered interaction network.

Although the *L2* and *L3* folds appear reminiscent of reported structures^{26,27}, fibril-lipid interactions favor novel quaternary protofilament arrangements. In the *L2A* fibril, lipid-mediated interactions seem to be essential as they connect the neighboring protofilaments. Lipid-mediated interactions might also be responsible for the protofilaments pointing in opposite directions in the *L2B* and *L3A* fibrils, as in this configuration, two mirrored ₃₄KEGVLYVGSK₄₃ segments from both protofilaments are in contact with the same phospholipid micelle (Fig. 3i, j). Again, the acyl chains bind to Y39, V40, and G41, the head groups interact with K34, K43, and K45 on both protofilaments, and Cl⁻ ions colocalize with head groups at the interface between K43 and K45 (Fig. 4).”

2. It seems that the structural model coordinates and the cryo-EM maps have not been submitted to PDB or EMDB. I strongly suggest the authors upload the files to these two databases and provide full wwPDB validation reports.

A: For the revision, we now uploaded the atomic models and the cryo-EM maps to PDB and EMDB, respectively. The accession codes are reported in the **Extended Data Tab. 1** on page 47. Please also find a summary below.

Lipid-induced PM	L1A	L1B	L1C	L2A	L2B	L3A
PDB-ID	8ADU	8ADV	8ADW	8A4L	8ADS	8AEX
EMDB-ID	15370	15371	15372	15148	15369	15388

3. Whether POPC/POPA vesicles affect the fibrillation kinetics of α -syn? The authors might need to perform ThT fluorescence assay to monitor the α -syn fibrillation kinetics in the presence of lipids.

A: The influence of POPC/POPA vesicles has been studied before in the same L/P range as presented here and showed an acceleration of α -synuclein aggregation (Jiang, de Messieres, and Lee, *J. Am. Chem. Soc.*, 2013). This was also reported for a multitude of other negatively charged phospholipids and our results show a similar trend. We added representative curves as Extended Data Fig. 1b (p. 27).

Extended Data Fig. 1b: Representative curves of normalized ThT fluorescence (I/I_{\max}) following the aggregation kinetics of α Syn in the presence (blue) and absence (magenta) of vesicles of POPA and POPC (1:1) under PMCA conditions. Curves were obtained by fitting the data to an unseeded secondary nucleation model using Amylofit⁶⁵ (www.amylofit.ch.cam.ac.uk). Lag-times t_{lag} were determined as the intersection of the x-axis and a linear function fitted to the steepest part of the curve

4. The authors might need to compare the L1 fibril fold with previously reported different α -syn folds, and discuss what's the novel structural feature of this fold.

A: Following the reviewer's suggestion, we extended the description of the novel L1 fold and compare the L1 fold to previously determined structures. Therefore, we added **Extended Data Fig. 4** in the revised version of the manuscript. The comparison starts on p. 4 at line 23 and reads: "The lipidic L1 fold reveals minor similarities to previously resolved structures of α Syn in the absence of phospholipids (**Extended Data Fig. 4a**). In detail, only the fold of the L1 segment V52 - T72 is found in the protofilament fold of wild type and Y39 phosphorylated α Syn (**Extended Data Fig. 4b, c**). This discrepancy with previously resolved structures is probably related to the presence of phospholipids during α Syn aggregation. While the previously determined structures are characterized by a predominantly hydrophobic core, in the L1 fold a surprisingly large number of hydrophobic residues are found on the surface (**Extended Data Fig. 4d**). However, these "solvent-exposed" areas are decorated with non-proteinaceous densities (**Fig. 1f-h**), corresponding to surface-bound phospholipids (for details, see below). Hence, the phospholipids may shield, at least to some extent, the hydrophobic amino acids on the fibril surface from direct interactions with water during α Syn aggregation, which then leads to the lipid-mediated L1 fold."

5. The distinct packing pattern of protofilaments in L2B and L3A fibrils is interesting and features a novel protofilamental packing symmetry. How did the authors determine the handedness of these fibrils? Atomic force microscope can be used to characterize the handedness of fibril twist. For high resolution maps, densities for the carbonyl oxygen atoms may also help to confirm the handedness.

A: We did not perform any additional experiment to investigate the handedness of the fibrils. However, **for all fibrils we found regions in the structure where the local resolution is sufficient to identify the orientation of the backbone carbonyl groups, and with this all fibrils were found to have a left-handed twist.** We have added this sentence to the Methods section.

6. The authors need to clarify how the relative population of each lipid- α -syn fibril polymorphs (**Extended Data Fig. 1g**) is determined.

A: We added an additional section (**Determination of the relative population of each fibril polymorph**) to the Methods part, which reads (p. 20): "In the cases of L1A, L1B, L1C, and L2A fibrils, the population relative to the total number of extracted helical segments was calculated based on the number of helical segments after the successful separation by 2D classification. As to L2B and L3A, on the other hand, we used the number of helical segments after successful separation by 3D classification."

7. As for the statement “In the *LIB* fibril, both protofilaments are related by an approximate 21 screw symmetry and the protofilaments are tilted by $\sim 37^\circ$ to each other.”, how is the tilt angle measured?

A: To estimate the tilt angle between the protofilaments, we measure the dihedral angle described by the C α atoms of M1-V40-M1'-V40' of two opposite protein chains. In the revised manuscript, please find the novel Extended Data Fig. 5 (p. 33) visualizing the tilted orientation of the protofilaments in *LIB*. For comparison, we also show *LIC* with an almost planar orientation of the protofilaments.

Minor concerns:

1. As for *LIC*, “ionic interactions between residues K45 and E46 form the inter-protofilament interface” should be “K43, K45 and E57”;

2. Twist angle of *L1A* in Figure 1b should be consistent with the number in Extended Data Table

3. Initial model used for model building of *L3A* should be 6UFR;

4. The tables cited in the main text is not in a right order.

A: We highly appreciate the careful reading. All concerns have been addressed in the revised manuscript.

Reviewer #2 (Remarks to the Author):

The manuscript describes cryo-EM structures of six α -synuclein fibril polymorphs in complex with lipids. The phospholipids are suggested to induced (some) new morphologies of the fibrils and to bind to cavities within them. The authors suggest that the structures support a mechanism of co-aggregation of lipids with fibril, and to fibril-induced lipid extraction, leading to cell toxicity and pathology via disruption of intracellular vesicles.

The “big question” here is whether the lipids really induce specific conformations that are relevant to membrane interactions or rather are just “additional polymorphs” of α -syn that are induced by different conditions or a part of an inherent population mix. I think it is impossible to answer this question with current tools. There are few residues that were correlated with lipid interactions, but this is a very limited information and support. The most convincing evidence is the presence of lipids in the central cavity of L2A that might mediate the interaction between the protofilaments. Having said that, the L2A protofilament arrangement was observed without lipids, then yet the intertwining of the protofilaments might be lipid induced and relevant. Really impossible to tell. I agree that it is tempting to suggest that the structures support a proposed lipid extraction mechanism and co-aggregation of lipids with fibrils.

1. Another conceptual question is the colloidal state of the lipids which encounter the protein/fibrils. Is it in the presence of vesicles or micelles (probably affected by the sonication/incubation)? Or some other type of colloidal system?

This might be relevant in case the interaction / state of the lipids after sonication is different in comparison to a vesicular system which resembles more physiological condition. Why didn't you use intact vesicles?

A: The main text did not make it obvious that we indeed started with intact small unilamellar vesicles (SUVs). We changed the text accordingly to avoid any confusion to the future reader. The new paragraph reads (p. 4, line 2): **“De novo aggregation in the presence of small unilamellar vesicles (SUVs) at a 5:1 lipid to protein ratio was induced by sonication under protein misfolding cyclic amplification conditions and completed under gentle orbital shaking to elongate the fibrils²⁰. SUVs consisted of a 1:1 mixture of 1-palmitoyl-2-oleoyl-sn-glycero-3-phosphate (POPA) and 1-palmitoyl-2-oleoyl-sn-glycero-3-phosphocholine (POPC) as a simplification of negatively charged synaptic vesicles²¹ to recapitulate the established binding of monomeric α Syn to lipids via its N-terminus^{22,23}. In agreement with previous studies we observed significantly reduced lag-times in the presence of these phospholipids²⁴.”**

Technical comments:

2. Fig 3 – “In a+d, the arrows highlight non-fibrillar densities” – should it be a-f? I think there is a mix-up in the colours of the map in the figure legend.

A: We appreciate the careful reading and fixed the mismatch in the revised manuscript.

3. The residues that were identified by ssNMR to bind to the PLs are not named explicitly except from within the image itself (3 hydrophobic residues? – what about the headgroups?).

A: (Please see the combined answer on comment #7)

4. Fig 3d,e – points to LIB and it is refer in the text discussing K6, K21, and E20 - but these are not shown.

A: (Please see the combined answer on comment #7)

5. Fig 3f – L1C fibril - K43, K45, and E57 should be shown in closeness to the Cl ion as discussed in the text.

A: (Please see the combined answer on comment #7)

6. Fig 3h - L2A - it is impossible to appreciate the real distance between the lysines and the Cl ions in this figure. “where Cl- ions colocalize with head groups at the interface between K43 and K45 (Fig. 3a,i,j)” – one cannot see it clearly in 3i,j and definitely not in 3a. zoom-in is needed.

A: (Please see the combined answer on comment #7)

7. Segments 35-EGVLYV-40 and 34-KEGVLYVGSK-45 should be indicated in the figures. Also a zoom-in of Y39.

A: In the revised manuscript, we now include a detailed structural analysis of the interactions between α Syn and POPA, POPC, Na^+ , and Cl^- throughout the MD simulations. To do so, we measured the minimal distance between any non-hydrogen atom of every amino acid of the central five layers of each protofilament to (i) the phosphate group of the phospholipids, (ii) the quaternary choline group of the phospholipids, (iii) any carbon atom of the acyl chains of the phospholipids, (iv) any Na^+ , and (v) any Cl^- ion. An interaction was present, if the distance was smaller 5 Å. These interactions are normalized by the total number of frames, so that a value of 1.0 means “interaction always present”, whereas a value of 0.0 means “interaction never existent”. We considered an amino acid as “interacting”, if the interaction is present in at least 50% (value 0.5) off all conformations and “strongly interacting” if the interaction is present in at least 75% (value 0.75) off all conformations. The results are shown in the new Fig. 4 (p. 15).

In addition, we extended the description and interpretation of the MD simulation data in the revised manuscript, which reads (pp. 6): “The patterns of lipid interactions per residue repeated in all LI fibrils suggest that lipid-mediated intramolecular interactions may be necessary for the novel LI folding. For all LI lipids, the predominantly hydrophobic segments $_{1}\text{MDVFM}_5$, $_{36}\text{GVLYV}_{40}$, $_{69}\text{AVVTGVTAVA}_{78}$, and $_{85}\text{AGSIAAATGFV}_{95}$ are in contact with the lipidic acyl chains. At the same time, the adjacent polar residues K6, E20, K21, K32, E35, N79, K80, and S87 interact with the lipidic head groups (Fig. 4). Hence, hydrophobic areas on the fibril surface are, at least partially,

covered with phospholipids. **Fig. 5** shows a POPC molecule modeled into the most well-defined non-proteinaceous densities at the fibril surface.

The central cavity in the *L1B* fibril is occupied by lipids, with their head groups bridging interactions between K6, K21, E20, and E35, while their acyl chains form hydrophobic interactions with M1, V2, M5, G36, L38, and V40 bridging across the protofilament interface (**Fig. 3d, e, Fig. 4**). The MD simulations revealed that the *L1B* cavity is occupied by chloride ions (Cl⁻), which are complexed by the positively charged residues K21 and K23. For the *L1C* fibril, also revealed a high probability for Cl⁻ ions in the hydrophilic interface involving residues K43, K45, and E57 is found (**Fig. 3f,g, Fig. 4**).

A striking feature of the *L2A* fibril is the bridging of lipid molecules that span the ~20 Å gap between the protofilaments. The simulations revealed that lipids interact with the segment ₃₃TKEGVLYVGSKTK₄₅, bridging the gap between the protofilaments. In detail, the acyl chains bind to Y39, V40, and G41, which form a small hydrophobic patch at the fibril surface (**Fig. 4**). Additionally, the lipidic head groups interact with K43 and K45 on one protofilament and with K34 on the neighboring protofilament (**Fig. 4**). The head group densities of these lipids partially overlap with densities for Cl⁻ (**Fig. 3h**) and the per-residue analysis confirmed that K34, K43, and K45 also interact with Cl⁻ (**Fig. 4**). Hence, the negatively charged phosphate groups and the Cl⁻ ions together form the bridge between K34 and K43 in the individual protofilaments by forming a well-ordered interaction network.

Although the *L2* and *L3* folds appear reminiscent of reported structures^{26,27}, fibril-lipid interactions favor novel quaternary protofilament arrangements. In the *L2A* fibril, lipid-mediated interactions seem to be essential as they connect the neighboring protofilaments. Lipid-mediated interactions might also be responsible for the protofilaments pointing in opposite directions in the *L2B* and *L3A* fibrils, as in this configuration, two mirrored ₃₄KEGVLYVGSK₄₃ segments from both protofilaments are in contact with the same phospholipid micelle (**Fig. 3i, j**). Again, the acyl chains bind to Y39, V40, and G41, the head groups interact with K34, K43, and K45 on both protofilaments, and Cl⁻ ions colocalize with head groups at the interface between K43 and K45 (**Fig. 4**).”

In contrast to the reviewer’s suggestion to show a structure from the MD ensemble with zoom-in onto the areas of interest, this approach provides an even more detailed picture about the lipid interactions, as it includes the ensemble information and not only one snapshot. We assume that such an analysis will also help the future reader to get a clear picture about the interactions of αSyn, lipids, and ions.

8. Please indicate the FSC=0.143 resolution in Extended Data Fig. 2 (crossing lines or something similar).

A: In the revised manuscript, we now show FSC=0.143 as gray line (see Extended Data Fig. 3 on p. 30).

9. I am confused by the 2.37Å helical rise of L1B. In the FSC there is a peak at ~4.7Å which is similar to the other polymorphs.

A: In the L1B fibril both protofilaments are related by an approximate 2_1 screw symmetry, leading to a staggered arrangement of the of the protofilaments relative to each other, which is also visualized in the close-up view in Fig. 1c. Still, within one protofilament the stacked α Syn peptides are separated by 4.74 Å. However, as the helical rise describes the spatial displacement of two asymmetric units along the helical axis, the rise between two staggered protofilaments yields 2.37 Å.

To avoid any confusion to the future reader, we modified Fig. 1c and now also show the rise per protofilament (= 4.74 Å) and the rise between two staggered protofilaments (= 2.37 Å).

10. In page 6 line 4 is stated: “The conversion of the SUVs used for the preparation of the lipidic fibrils to such small lipid aggregates upon fibril formation was confirmed by ^31P ssNMR”, so it means that the SUVs were used as initial system? Is it the same lipid state as used for the cryo-EM?

A: Yes, SUVs were used to during α Syn aggregation. As stated above (please see comment 1), we now explicitly mention the SUVs in the main text.

11. Why are the PDB reports “*Not For Manuscript Review? I have never seen this statement. There are many clashes, but I think mostly of hydrogens, so it is fine. Anyway, the resolution seems to be good enough to avoid clashes.

A: We have now deposited the atomic models and the cryo-EM maps to PDB and EMDB, respectively. The accession codes are reported in the **Extended Data Tab. 1** on page 47. Please also find a summary below. The number of clashes is slightly higher than average observed for EM structures of similar resolution, however we think the number of clashes is still in an acceptable range.

Lipid-induced PM	L1A	L1B	L1C	L2A	L2B	L3A
PDB-ID	8ADU	8ADV	8ADW	8A4L	8ADS	8AEX
EMDB-ID	15370	15371	15372	15148	15369	15388

Reviewer #3 (Remarks to the Author):

The authors of the study present 6 new cryo-electron microscopy α -synuclein fibril structures. They also support their experimental work with MD simulations. The study is well-structured, up-to-date and original, and provides new insights on lipid- α -synuclein fibril interactions that could guide future therapeutic applications for PD. However, authors should address the following questions before work is accepted

- Although the authors gave a reference, how many ns did it reach thermalization and density adaptation in MD simulations? this time period should be included in the manuscript. Also, is it possible to embedded a visual proof of these metrics on the supporting information?

A: (please see next comment)

- In which statistical ensemble did the authors perform their equilibration simulations? this must be stated in the manuscript.

A: (please see next comment)

- It was interesting that the MD replicas were produced at the start of the NPT production simulations. Could the replicas have been produced at the start of the balancing simulations? Can the authors review this situation? Also, how were the initial velocities for these replicas randomly determined? Were they produced based on any physical function?

A: Following the reviewers suggestions, we now show the thermalization and density adaptation data in the new **Extended Data Fig. 9** (p. 42). Additionally, the figure caption also includes a description of the simulation procedure. The new caption (starting on p. 43) reads: “Time series of the temperature (left panels) and density (right panels) over 0.6 ns of MD simulations for *L1A* (a), *L1B* (b), *L1C* (c), *L2A* (d), *L2B* (e), and *L3A* (f). The MD simulation procedure⁵⁸ started by heating the systems from 0 K to 100 K in a canonical (NVT) MD simulation of 50 ps length. Afterward, the temperature was raised from 100 K to 300 K during 50 ps of isobaric-isothermal (NPT) MD. Subsequently, the density was gently adjusted to 1 g/ml during 200 ps of NPT-MD. During the heating and density adaptation steps, positional restraints of $1 \text{ kcal}\cdot\text{mol}^{-1}\cdot\text{\AA}^{-2}$ were applied to all backbone and lipidic phosphate atoms. These harmonic positional restraints were removed from the lipidic phosphate atoms by gradually decreasing the force constant from 1 to 0 $\text{kcal}\cdot\text{mol}^{-1}\cdot\text{\AA}^{-2}$ in six NPT-MD runs of 50 ps length each. From here, we (re-)started eight independent NPT production simulations at 300 K and 1 bar for 1 μs each, in that new velocities were assigned from Maxwell-Boltzmann distribution. Importantly, we restrained the backbone to the initial atomic coordinates throughout production simulations while all other atoms were allowed to move freely.”

-The authors say that after 100 ns, the simulations convergence. Can they justify this situation with various visual metrics and put them in supporting information?

A: We more specifically only claimed that the distribution of lipids converge after 100 ns, we wrote: “Thereby, we observed only minimal changes when extending the analysis time from 0.9 μs ns to 1.0 μs , such that we assumed converged distributions of the lipid molecules.”. However, we agree with the reviewer that some visual proof may increase the understanding and in the revised

manuscript we now show the progression of the density grids throughout one replica simulation for all six fibril structure in the new **Extended Data Fig. 11** (p. 45). Although we decided to focus on the distribution of the lipids and do not show the density grids of ions for clarity purposes, the **Extended Data Fig. 11** visualizes that, first, major changes in lipid distribution appear on the first 0.3 μs , second, the changes in the lipid distribution are minimal when extending the analysis time from 0.6 μs to 0.7 μs , and, finally, the changes in the lipid distribution are neglectable when extending the analysis time from 0.9 μs to 1.0 μs . Hence, we assumed converged distributions of the lipid molecules.

-In the simulation video, it looks like position constraints have been applied to the fibrils. If so, the authors should definitely state this situation in the manuscript along with their reasons.

A: The fibril's backbone was restraint to the initial coordinates and we provided this information in the original manuscript in the Methods section: "*Importantly, we restrained the backbone to the initial atomic coordinates. However, all other molecules, including POPC and POPA, were allowed to diffuse freely and we did not apply any artificial guiding force.*". As suggested, we extended this section by a short explanation for our reason, including the novel **Extended Data Fig. 10** (p. 44), which shows RMSD plots of the fibril structures after 1 μs MD simulations without lipids and without positional restraints. We have to restrain the backbone because the fibrils are not stable without the proper converged lipid distribution around the fibril, which was not present at the beginning of the simulations. The **Extended Data Fig. 10** visualizes that the quaternary structures of the models are not stable without positional restraints.

Reviewer #4 (Remarks to the Author):

The authors present structures of alpha-synuclein fibrils prepared from in vitro incubation with phospholipids. This is highly significant work because it presents the first direct structural evidence of lipid interactions with synuclein fibrils and the structures are good quality.

Some drawbacks of the study are that the lipid to protein ratios are unusually low and there are large differences among the various structures observed, so it is not clear whether these actually represent situations observed physiologically. This criticism, however, is true for most alpha-synuclein structural studies, so it is not a fatal flaw.

The structures are well described and complementary MD simulations and solid state NMR studies support the overall impact of the work.

The validation reports indicate that they are not intended for submission to manuscripts so this seems awkward.

A: For resubmission, we uploaded the atomic models and the cryo-EM maps to PDB and EMDB, respectively, including final validation reports. The accession codes are reported in the **Extended Data Tab. 1** on page 47. Please also find a summary below.

Lipid-induced PM	L1A	L1B	L1C	L2A	L2B	L3A
PDB-ID	8ADU	8ADV	8ADW	8A4L	8ADS	8AEX
EMDB-ID	15370	15371	15372	15148	15369	15388

REVIEWERS' COMMENTS

Reviewer #1 (Remarks to the Author):

The authors addressed my concerns with satisfaction. I support publication of this work in NC.

Reviewer #2 (Remarks to the Author):

The authors have satisfactorily addressed my comments.

Reviewer #3 (Remarks to the Author):

The authors have made significant changes as per suggestions and have increased the quality of work and readability. I think manuscript can be considered for the publication.